# Environmental Enrichment for Sucker and Weaner Pigs: The Effect of Enrichment Block Shape on the Behavioural Interaction by Pigs with the Blocks

**DOI:** 10.3390/ani7120091

**Published:** 2017-11-27

**Authors:** Jade A. Winfield, Greg F. Macnamara, Ben L. F. Macnamara, Evelyn J. S. Hall, Cameron R. Ralph, Cormac J. O’Shea, Greg M. Cronin

**Affiliations:** 1School of Life and Environmental Sciences, The University of Sydney, 425 Werombi Road, Camden NSW 2570, Australia; jade.winfield@gmail.com (J.A.W.); gregory.macnamara@sydney.edu.au (G.F.M.); benjamin.macnamara@sydney.edu.au (B.L.F.M.); cormac.oshea@sydney.edu.au or Cormac.O’Shea@nottingham.ac.uk (C.J.O.); 2Biometrics Group, Sydney School of Veterinary Science, The University of Sydney, Camden NSW 2570, Australia; evelyn.hall@sydney.edu.au; 3South Australian Research and Development Institute, Animal Welfare Science Centre, The University of Adelaide, Roseworthy Campus, Roseworthy SA 5371, Australia; cameron.ralph@sa.gov.au

**Keywords:** enrichment, pigs, production

## Abstract

**Simple Summary:**

How often do intensively-housed pigs interact with enrichment devices, especially smaller pigs during the sucker and weaner phases of production? Understanding whether smaller pigs use such devices, whether habituation occurs, and whether device-shape influences the level of interaction by pigs are all relevant questions surrounding justification for including enrichment devices in pig pens. We provided litters of pigs from 10 to 60 days old with one of three different shaped enrichment blocks (cube, brick, or wedge) and recorded oro-nasal contact with the blocks on two days each week. Fresh blocks were provided weekly. While pig interaction with blocks was infrequent before about 25 days of age, contact with the blocks steadily increased thereafter. Brick-shaped blocks were used more than the other shapes, possibly because the brick-shape presented a wider surface for contact, allowing multiple pigs to simultaneously interact with the block. Pigs interacted more with 1-day-old (i.e., ‘fresh’) than 4-day-old blocks, suggesting that habituation to the blocks may have occurred.

**Abstract:**

This experiment tested the effect of enrichment-block shape on oro-nasal contact by young pigs, and possible habituation to the blocks. Nineteen litters (197 piglets) were randomly allocated to one of three block-shape treatments: Cube, Brick, or Wedge. Oro-nasal contact with blocks was infrequent before 25 days of age. Thereafter, contact steadily increased, suggesting enrichment blocks may not need to be provided until week 4 of lactation. Brick-shaped blocks attracted more oro-nasal contact than the cube and wedge shapes (*p* = 0.002). Oro-nasal contact was more frequent (*p* < 0.001) during the first 24 h after block introduction than when blocks were four days old. From 25 to 60 days of age, oro-nasal bouts were longer (*p* = 0.014) during the first 30 min of exposure to a fresh block, than for the remainder of the 24 h, or on day 4 after block replacement. Therefore, habituation to blocks may have occurred by 24 h after block introduction. Brick-shaped blocks may present a wider surface for oro-nasal contact, where multiple pigs could simultaneously interact with the block. We speculate that simultaneous interaction with brick-shaped blocks may be similar to a litter co-operatively massaging the sow’s udder prior to suckling bouts.

## 1. Introduction

Environmental enrichment is provided to captive zoo animals, as well as companion, laboratory, and farm animals to assist in improving welfare by increasing the frequency and diversity of (natural) behaviours performed [1,2,3]. There is public expectation that intensively farmed animals such as pigs should also receive environmental enrichment to ensure sufficient species-specific behaviour is performed to allow pigs to enhance their behavioural repertoire [4,5]. In some countries this is also a legislative requirement. In order to maximise welfare outcomes, young pigs need to be housed in groups enabling social interaction, and with sufficient space and enrichment materials to stimulate normal exploratory behaviours [6]. Indeed, it has been concluded that commercial pigs require inanimate objects for “proper investigation and manipulation” to ensure good welfare [4]. Thus, social living combined with the provision of extra space do not necessarily ensure good welfare status for pigs, for example, if the pigs perceive the environment to be “barren”. Pigs require manipulable objects or bedding for stimulation [7]. However, under commercial conditions in Australia, bedding is problematic since the conventional indoor, pig-pen environment is not compatible with straw or other bedding material. Straw may block drains, pipes, and pumps located under the slatted flooring, resulting in effluent collection systems failing [8]. Nevertheless, the 2006 review by Bracke et al. [4] and the subsequent European Community Directive (Directive 2008/120/EC) [9] conclude that there should be permanent provision of material such as straw to pigs. While the characteristics of alternative, inanimate objects pigs find to be enriching as ‘ingestible’, ‘odorous’, ‘chewable’, ‘deformable’, and ‘destructible’ have been listed [10], it is also recognised that pigs may quickly lose interest in (that is, habituate to) inanimate objects in their pen.

The literature is not conclusive that the provision of enrichment objects necessarily results in improved welfare of sucker/weaner pigs, for example, compared to the provision of a chain to chew [11]. Nevertheless, many studies have focused on increasing the diversity of species-specific behaviours, as some believe this contributes to improved welfare of animals [12]. Previous studies on pigs have concentrated on foraging as an important species-specific behaviour [13]. Habituation is less likely to occur if the enrichment object motivates the animal to perform behaviours that are relevant to survival, such as ingesting food or avoiding predators. Hence, pigs may quickly lose interest in enrichment devices provided in their pen if the devices are not “relevant” [10]. Key questions therefore include how to ensure pigs interact with enrichment devices provided, and how to stimulate re-investigation or continued use of enrichment devices?

Food is an important motivator for pigs, often being more effective than social interaction [14]. Foraging is a series of appetitive behaviours whereby pigs investigate the environment with the goal of seeking food to ingest, which presumably rewards them for their behaviour. Providing an occasional food reward would seem to be important when attempting to reinforce foraging behaviour in an animal which is directed to an inanimate “enrichment” object. The reward encourages the animal to interact with the object again, which decreases the chances of habituation [10]. In Australia, a “nutritional lick block” has been developed by Ridley Agriproducts (Toowong, Queensland, Australia) for adult sow enrichment in the gestation housing phase of commercial production [15]. Ingestion of the enrichment block material by pigs potentially provides a reward to reinforce foraging behaviour towards the block. In theory, ingestion of (edible) block material should motivate sows to continue to interact with the blocks, preventing habituation. Currently, these hard blocks are produced for adult sows and are poured as a cube measuring about 26 × 26 × 26 cm and weighing ~20 kg. It is not known whether smaller pigs, such as suckers and weaners, would be able to root/bite/chew such heavy cube-shaped blocks. If enrichment blocks are provided to smaller pigs that are incapable of manipulating them due to the large size and mass, or due to inappropriate shape, it is a risk that pigs may habituate to the blocks when provided in their pens and there will be no potential benefit to welfare. In this experiment we aimed to investigate the effects of block shape on the occurrence of oro-nasal contact by sucker and weaner pigs with hard blocks, and to determine whether habituation to the blocks occurred, for example, if pigs could not root, chew, or bite the blocks, or were otherwise not stimulated to interact with the blocks. We hypothesised that oro-nasal contact with blocks would be increased firstly, if smaller-sized pigs could bite or chew the block, and secondly, if facilitative (co-operative) group behaviour could be stimulated.

## 2. Materials and Methods

### 2.1. Litters and Housing

The experiment was conducted at the University of Sydney piggery, Camden, under approval of the University of Sydney Animal Ethics Committee (approval number: 2016/968). A total of 197 piglets from 19 litters were used. The dams were crossbred Large White-Landrace sows, and litter size (total born) at farrowing ranged from 3–19 piglets. Mean (±std dev) sow parity number was 2.9 ± 1.59 litters, and born alive per litter was 11.5 ± 3.67 piglets. At about 24–36 h postpartum, piglets were weighed and underwent ear-notching, tail docking (last one-third of tail), and teeth clipping in line with the husbandry practices on the farm. Piglets also received an intramuscular iron injection. Litters were randomly assigned to treatments, and the experiment was conducted over three replicates in time corresponding to farrowing batches in the pig unit. Litters were not mixed during the experiment. During the sucker stage, litters were housed with their dam in conventional farrowing crates in an environment-controlled room containing eight farrowing crates (measuring 1.6 × 2.2 m with fully-slatted, tri-bar metal flooring). An area of solid flooring (0.5 × 1.8 m) was provided for the piglets in the heated creep area. A small amount (handful per crate) of creep feed was provided daily on the floor of the creep area from day 7, and water was available ad libitum via a piglet nipple drinker. At weaning the litters were transferred to an adjacent, heated weaner room containing eight raised pens (2.35 × 1.2 m). The weaner pens had 1.55 × 1.2 m of solid flooring and 0.8 × 1.2 m of tri-bar slatted flooring. Pelleted weaner diet and water were provided ad libitum in the weaner pens.

### 2.2. Enrichment Block Shape Treatments

Full-sized enrichment blocks were obtained from Ridley Agriproducts and cut into three shapes: a cube, a brick, and a wedge, as represented schematically in Figure 1. Each litter in their crate or weaner pen was only exposed to one block shape (treatment) throughout the experiment. Relevant dimensions and angles of the three block shapes for sucker and weaner pigs are presented in Table 1.

The logic of comparing pig responses to the three block shapes were as follows:
(1)The cube would potentially provide enough space for a single sucker or weaner pig only to interact with one surface of the block, and adjacent surfaces would be separated by right angles that piglets might find difficult to mouth or bite. Thus, the cube shape served as a control treatment for the other shapes.(2)The brick shape might enable multiple pigs to interact with the block, for example, in a co-operative manner similar to how littermates might massage the sow’s udder. Thus, the brick shaped block served as a treatment that potentially allowed facilitation of co-operative group behaviour.(3)The wedge shape had two edges of 45 degrees, which we assumed would enable smaller pigs to mouth or bite into the block (compared to a right-angle edge). Thus, the wedge-shaped block served as a treatment that potentially allowed especially piglets to more easily taste and/or ingest block material through mouthing/biting the narrow edge, thereby rewarding oro-nasal contact with the block through taste or consumption of the material.

### 2.3. Presentation of Treatment Block Shapes to the Sucker and Weaner Pigs

Each shaped block was drilled through using a 10-mm diameter drill bit, which enabled the block to be skewered on a 10-mm-thick metal rod (see Figure 1). Thus, the blocks could be fixed on the vertical rods within farrowing crates/weaner pens to facilitate observation via video cameras. In addition, the shaped blocks were raised 20 mm above the slatted floor via a spacer located under the block, through which the metal rod passed. In farrowing crates, the attachment rod skewering blocks was positioned about 20–25 cm from the front and (creep-) side wall of the farrowing crate, and about 35–40 cm from the sow stall. In weaner pens the attachment rod was positioned centrally, about 25–30 cm from the rear wall. Blocks were positioned above slatted rather than solid flooring to facilitate attachment of the lower end of the vertical metal rod from below the floor.

Blocks were weighed (after drilling out the hole for the rod) prior to placement and were removed and re-weighed after 7 days to determine disappearance rate. Each week a new (fresh) block of the same appropriate treatment shape was placed in each crate/pen. In the event that the block was consumed within the 7 days, a new block of the same shape was prepared, weighed, and placed in the pen. The initial weights of the different shaped blocks provided to the sucker pigs averaged 0.89 kg, 1.71 kg, and 0.48 kg for the cube, brick, and wedge, respectively. When the piglets reached the weaner phase, larger block sizes were used, with mean weights of 2.79 kg, 5.31 kg, and 1.40 kg, respectively, for the cube, brick, and wedge treatment shapes. Weaning occurred on a set day of the week (Thursday) in the fourth week of lactation. Average weaning age was about 26 days.

The treatment blocks were first introduced to the sucker pigs in farrowing crates when the litters averaged 10.7 days old (±2.13 std dev). Replicate 1 concluded when the weaner pigs averaged 66 days old (±1.99 std dev). However, due to the wedge-shaped blocks being “destroyed” daily by some litters of weaner pigs in replicate 1, once the litters were about 40 days old, it was decided to conclude the observations in the subsequent two replicates once litters were about 7 weeks old (average 46.5 days ± 1.92 (std dev) for the second and third replicates). The number of litters receiving the three treatment block shapes per replicate is shown in Table 2.

### 2.4. Pig Growth Measurement

All pigs were individually identified using ear notching that had been performed when the piglets were 2 days old. During the experiment pigs were weighed weekly, coinciding with weighing of the enrichment blocks. While pig live weight data were collected to investigate the effects of block shape on growth of the litters, weight gain data were also monitored to determine whether growth was associated with block disappearance within weeks.

### 2.5. Video Recording

All farrowing crates and weaner pens were monitored using low-light video cameras (AHDI Mega Pixel Cameras, CCTV Central, Mount Waverley, Victoria, Australia) with 3.6 mm fixed lenses. Cameras were mounted above the respective locations to provide a clear view of the enrichment block. Sound was not recorded. Video data were continuously recorded using a 16 Channel Analogue High Definition (AHD) 1080P Digital Video Recorder (DVR) (CCTV Central, Mount Waverley, Victoria, Australia). Digital video data were then downloaded from the DVR unit to external hard drives in the format of Audio-Video Interleave (AVI) files, which were viewed on a laptop computer for collation of behaviour data. Lights were left on in the farrowing and weaner rooms at night to assist recognition of focal pigs on the video record. This was necessary to help avoid over-exposure of pigs on the video record, whilst also improving illumination of the blocks. The blocks were relatively dark in colour and the pig heaters emitted bright (visible) light that could cause over-exposure within the field of view reducing the observer’s ability to identify focal pigs.

### 2.6. Behaviour Observations: 1—Duration (Bout Length) of Interaction

In each litter, four piglets were randomly chosen (two males and two females) to be focal pigs. Focal pigs were marked with a number and/or a colour on the back for easy recognition on the continuous video record. Identifying individual (focal) pigs on the video record enabled the quantification of the duration of oro-nasal contact (i.e., bout length) by specific pigs with the block shape in their pen. Data were collated from the video record during three time periods per week, related to the time of introduction of the fresh enrichment block:
The first 30 min after placement of a fresh block (First 30 min)The first 5 min per hour over the next 23 h (Day 1)The first 5 min per hour on the fourth day after placement of the block (Day 4)

### 2.7. Behaviour Observations: 2—Frequency of Interaction

The number of pigs (focal or non-focal) per litter performing oro-nasal contact with the treatment block was recorded using a point sampling technique [16] at predetermined time points from the video record to provide a quantitative measure of the frequency of oro-nasal interaction with the blocks:
On each minute over the first 30 min after placement of a fresh block (First 30 min)On each minute over the first 5 min/h over the next 23 h (Day 1)On each minute over the first 5 min/h on the fourth day after placement of the block (Day 4)

The Day 4 data were assumed to assist in determining whether habituation to the blocks occurred, that is, compared to the first 30 min after fresh blocks were provided, or on Day 1. The frequency data also permitted assessment of the “popularity” of the three block shapes, by counting the number of pigs performing oro-nasal contact with each block shape at each time point. A pig was considered to perform oro-nasal contact with the block if the snout was close enough to the block for rooting, biting, or chewing to occur, and there were no obvious signs of inactivity such as the pig was sleeping, fighting with another pig, or biting the metal rod rather than the enrichment block. If the observer was uncertain whether oro-nasal contact had occurred between the pig and the block, the video would be continuously watched for up to 5 s after the observation point to clarify the pig’s actions.

### 2.8. Statistical Analysis

Focal pig data were used to estimate the duration of oro-nasal contact by individual pigs with the block (i.e., bout length). Bout length was analysed using a Poisson model and a Generalised Linear Mixed Model (GLMM) analysis (Genstat ver 17.1, VSN International Ltd, Hemel Hempstead, UK). The data for the number of pigs performing oro-nasal contact with the block were converted to a binary system with 0 representing no contact by any pig(s) at the observation time point, and 1 representing one or more pigs were contacting the block at that observation point. The binary data were then analysed using a GLMM analysis in Genstat (ver 17.1). The findings for frequency of oro-nasal contact are therefore presented as the probability that pigs were contacting the block when the observation occurred. The correlation between piglet growth (live weight increase) and block disappearance was tested using a Restricted Maximum Likelihood (REML) variance components analysis in Genstat (ver 17.1). Back-transformed means (btm) are presented in the tables and figures for interpretation of probabilities.

## 3. Results

### 3.1. Animal Numbers, Weights, and Space Allowance during the Experiment

The experiment commenced when the different shaped enrichment blocks were introduced to the litters in the farrowing crates. As explained above in Section 2.3, for replicates 2 and 3 the trial period ended when the pigs were approaching 7 weeks old. Details of litter size and live weight of the pigs as they progressed through the trial are presented in Table 3. It should be noted that in Table 3 the full data set corresponding to all 19 litters (three replicates) is limited to weeks 1–5 of the trial. The space allowance at week 8 on trial for the group with the heaviest mean weight (20.46 kg/pig, Rep. 1, litter 8) and largest group at week 8 (*n* = 13 pigs, Rep. 1, litters 3 and 6, 16.92 and 17.18 kg, respectively) were always within the Australian Pig Welfare Code minimum space allowance requirements. The minimum space allowance for 21 kg pigs is 0.23 m^2^.

### 3.2. Effect of Block Shape on Frequency of Oro-Nasal Contact by Pigs with the Blocks

Averaged across observation sessions within and between weeks, the probability that a pig was performing oro-nasal contact with the block when observations were collated from the video record was 31% greater for the brick shape compared to the other shapes (*p* = 0.002). The probability of contact with the brick-shaped block was 17.0% (btm), compared to 13.2% and 12.7% for the cube- and wedge-shaped blocks, respectively. The likelihood that pigs performed oro-nasal contact with the different shaped blocks during each of the three time periods per week, respectively, is shown in Figure 2a–c, across the weeks of the experiment. In total, during the study, about 30,000 observations were collated from the video records. For the Cube, Brick, and Wedge block-shapes, respectively, a single pig was recorded interacting with a block on 8.4%, 10.1%, and 8.1% of occasions, while two or more pigs interacted on 6.4%, 7.3%, and 5.8% of occasions. Thus, during the majority of observations (~85%) pigs did not interact with the blocks.

The frequency of oro-nasal contact with the blocks was strongly influenced by age of the litter (*p* < 0.001). However, there were statistical interactions between litter age and enrichment block “age” (*p* < 0.001). The relationship between these factors is represented in Figure 3. Pigs were more likely to be observed performing oro-nasal contact with a “fresh” block (i.e., on Day 1) than the same block when it was four days old. Further, as litters grew older the slope of the line representing the level of oro-nasal contact with a “fresh” block (i.e., on Day 1) was slightly steeper than that for a four-day-old block.

### 3.3. Effect of Block Shape on Duration of Oro-Nasal Contact by Pigs with the Blocks

The duration of contact with blocks (bout length) by the focal pigs was not affected by block shape (btm: 1.7, 1.7, and 1.5 min, respectively for the cube, brick, and wedge shape; *p* = 0.580). Mean bout length, however, was significantly longer (*p* < 0.001) during the first 30 min of access to a fresh block (2.9 min), compared to the first 24 h (1.5 min) or on the fourth day (1.1 min) that the block was in the pen, and there was a significant interaction (*p* = 0.014) between the “freshness” of the block and age of the litters, as represented in Figure 4.

### 3.4. Change in Enrichment Block Disappearance Compared to Pig Growth

There were no effects of block shape on disappearance rate of the blocks within weeks (*p* = 0.347), that is, the residual weight of the block divided by the starting weight expressed as a percentage of weight loss. The mean loss in weight (i.e., disappearance of block material) expressed as a proportion of initial weight was 52%, 51%, and 60% for the cube, brick, and wedge shape treatments, respectively. Relative weight change (loss) of the different shaped blocks over the course of the trial is shown in Figure 5. There were significant block shape × week interactions (*p* < 0.001).

While relative block disappearance is a useful measure to describe the rate of disappearance of the enrichment blocks, a more relevant variable might be the absolute block disappearance, since this may more accurately reflect consumption of the block material by the litter of pigs. After adjusting for number of pigs in the litter, there was a difference in absolute block disappearance of the blocks over the course of the experiment (Figure 6). Significantly more material disappeared from the brick-shaped blocks than from the cube- or wedge-shaped blocks (mean weight loss values from the brick, cube, and wedge treatments were 2.2, 1.5, and 1.1 kg per week, respectively; *p* = 0.045, average sed 0.3551).

The data for absolute block disappearance change in the enrichment block (per pen) compared to growth of the pigs, after adjusting for litter size, were modelled in REML. There was no significant association between pig weight gain and block disappearance per week of the experiment (*p* = 0.322).

## 4. Discussion

Two hypotheses relating to increased oro-nasal contact by pigs with the enrichment blocks were proposed in this experiment. Oro-nasal contact with blocks would be increased firstly, if smaller-sized pigs could bite or chew the block, and secondly, if facilitative (co-operative) group behaviour could be stimulated.

We predicted that the wedge-shaped block would receive more oro-nasal contact from the pigs than the cube, as the 45-degree angled edges of the wedge should make the object more easily manipulable, especially by young sucker pigs. Oro-nasal contact should provide opportunity for positive feedback to the pig performing the chewing or biting, especially if a reward could be obtained. However, there was no significant difference between the probability of contact with the two shapes, suggesting that the hypothesis was not supported by the findings.

The brick-shaped block, however, was more ‘popular’ than either the cube- or wedge-shaped blocks. The probability of oro-nasal contact was 31% greater for the brick compared to the cube or wedge shapes, suggesting that the hypothesis was supported. This finding may be related to the physical size of the brick block, in which the dimension of one surface of the brick was wider than for the other shapes (see Figure 1). In theory, the brick-shaped block could enable multiple piglets to stand shoulder-to-shoulder to perform the typical behavioural pattern of piglets attempting to massage the sow’s udder and stimulate nursing behaviour in the sow. It has been reported that multiple piglets need to be present, co-operatively massaging the udder [17,18] for a suckling bout to be successful and result in milk ejection. Thus, the greater level of contact with the brick may have been related to the width of the block, which facilitated multiple pigs rooting, nosing, or biting the block simultaneously. Alternatively, as the brick was larger in size than the other shapes, more pigs could have simultaneously interacted with it. Hence, some caution is required here since the different shaped blocks were neither equal in volume nor mass.

A key finding from this experiment was that there was relatively little oro-nasal contact with any of the blocks until the litters were between 18 to 25 days of age (see Figure 2 and Figure 3). Alternatively, it is possible that at younger ages piglets perceived the type of material in the block, or the block shapes, to have low relevance. Nevertheless, younger pigs did not seem to be as ‘attracted’ to the blocks as older pigs. The finding may be associated with changes in sow milk production, or even with weaning per se which occurred on average around 28 days of age. Sows reach peak milk production by about day 21 of lactation [19], and thereafter milk output does not increase, whereas once piglets are weaned sow milk is no longer available and the weaner pigs need to consume an alternative form of feed. Regardless, after peak milk production is reached or weaning occurs, appetite would be expected to continue to increase as the pigs grow. The general increase in oro-nasal contact with the blocks as the pigs grew may have been a result of increased foraging behaviour. Foraging is relevant to survival, in that animals are motivated to seek food [20]. In the farrowing/lactation environment, although creep feed was supplied, it was supplied in relatively small quantities each day and hence the main source of food for sucker pigs was the sow’s milk. Presumably, as lactation progresses towards weaning the gap between appetite of the sucker pigs and the amount of milk available from the sow increases, motivating piglets to perform increased levels of foraging behaviour. The lack of oro-nasal contact in the earlier weeks of the trial, however, suggests that the ideal time to introduce blocks may be between day 18 and day 25 of age. While the frequency of pigs contacting the blocks increased with pig age (see Figure 3), within weeks the performance of oro-nasal behaviour directed at the blocks was also significantly greater in the first 24 h after fresh blocks were provided, compared to 4 days later and, is suggestive of habituation to the blocks. Potentially, the fact that blocks were replaced weekly may have slowed habituation. Nevertheless, on the fourth day after fresh blocks were provided there appears to be some evidence of habituation (see Figure 3). While it has been reported that the regular replacement or rotation of different enrichment objects did not prevent habituation, it slowed the rate of habituation [21]. Figure 4 may also be useful to estimate whether pigs had habituated to the blocks within weeks of age, by visually comparing the duration of oro-nasal contact by pigs with the blocks between the initial 30 min (first 30 min), the first 24 h (Day 1), and for a 24-h period 4 days after the block entered the enclosure (Day 4).

In the present experiment mean bout length (duration) of oro-nasal contact with blocks by the focal pigs was longer when the blocks were fresh. This was especially noticeable in the initial 30 min compared to the first 24 h after exposure, and on the fourth day after exposure. Mean bout length during the initial 30 min period after exposure to blocks was more than double that recorded for the remainder of the first day or on the fourth day, in all weeks after day 25 of age. This suggests that while fresh blocks provided some degree of novelty, pigs habituated to blocks possibly as soon as 24 h after exposure (Figure 4). Based on the present experiment, in which a fresh block was provided each week, it could be concluded that the enrichment blocks provided less stimulation, or were less enriching, compared to the enrichment that sustained piglet interest for 5 days in one other reported study [10]. Compared to the multifaceted enrichment devices previously used by Van de Weerd et al. [10], the blocks used in the present study are more practical and economic.

The enrichment block weights and pig weights were recorded at the beginning of each week so that associations between block disappearance and pig growth (weight gain) could be identified. As the pigs aged, the amount of block degradation per week typically increased. Presumably much of the weight loss from the blocks was due to ingestion of the material, and although some litters of weaner pigs in this trial exhibited scours in weeks 7 or 8 of the experiment, there was no apparent decline in growth rate. A particular issue occurred in the last few weeks of replicate 1, with the wedge-shaped blocks being prone to complete destruction by the weaner pigs. Accordingly, new blocks were replaced in the pens, sometimes daily. A limitation of this study, therefore, was that the different shaped blocks (treatments) were not isovolumetric, and the wedge-shaped blocks were volumetrically smaller, and of lower mass, than the other treatments. To avoid the problem of rapid destruction of the wedge blocks in replicates 2 and 3, we decided to conclude those replicates sooner than replicate 1. Thus, behavioural responses indicative of habituation in the wedge-shaped treatment in weeks 6–8 on experiment refer only to responses measured in replicate 1. Therefore, caution is required since disappearance rate of the wedge block may be confounded by block age and not an effect of the block treatment per se, although the wedge shape, and lower volume and mass, probably contributed to the pigs’ ability to (learn to) destroy that block shape faster. Similarly, we acknowledge that this was a relatively small study and a larger study with more litters is justified. Nevertheless, with more than 30,000 observed time-points included for this observational study, useful information is presented to assist in the design of enrichment blocks for smaller pigs.

## 5. Conclusions

While a brick-shaped enrichment block was more attractive to sucker and weaner pigs than a cube- or wedge-shaped block, it was probably not necessary to provide the blocks to pigs until at least 25 days of age.

## Figures and Tables

**Figure 1 animals-07-00091-f001:**
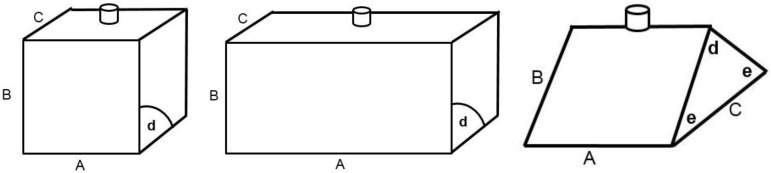
Sketches of the three block shapes used in the experiment. From left to right: cube, brick, and wedge, respectively. The position of the attachment skewer, using a 10-mm diameter metal pin inserted vertically through the block, is also shown in the diagrams. The different letters shown with the shapes refer to respective measurements of side length (**upper case**) or corner angles (**lower case**). The measurements are provided in Table 1 for each block shape prepared for sucker compared to weaner pigs.

**Figure 2 animals-07-00091-f002:**
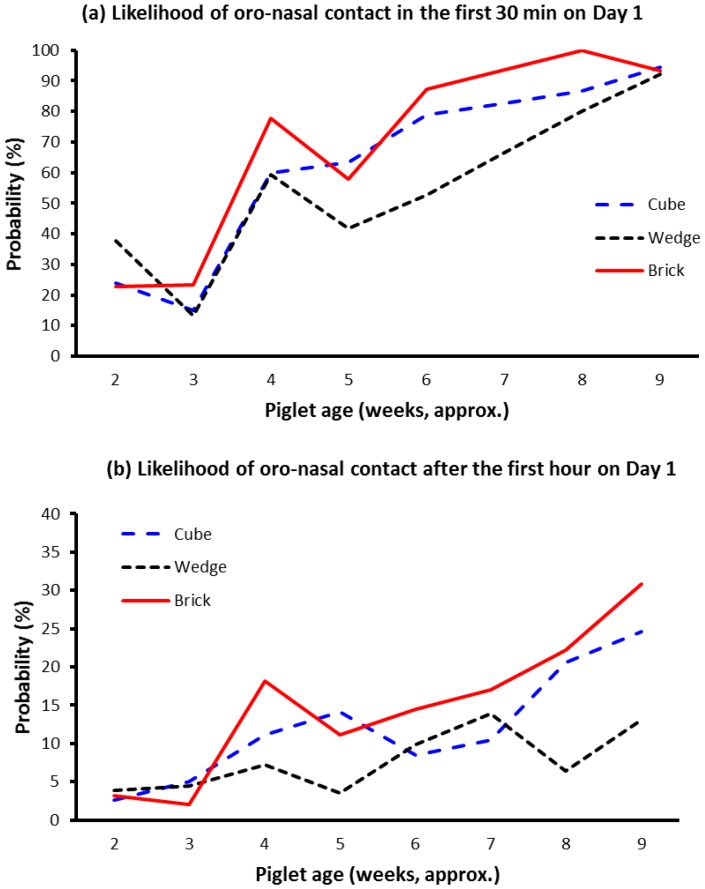
Oro-nasal contact by pigs with enrichment blocks of the three shapes during (**a**) the first 30 min after replacement in the pen, (**b**) during the next 23 h (Day 1), or (**c**) on the fourth day (Day 4). New blocks were provided each week. Values shown are the raw mean probability per treatment (shape) over time. Pigs were weaned at about age 28 days.

**Figure 3 animals-07-00091-f003:**
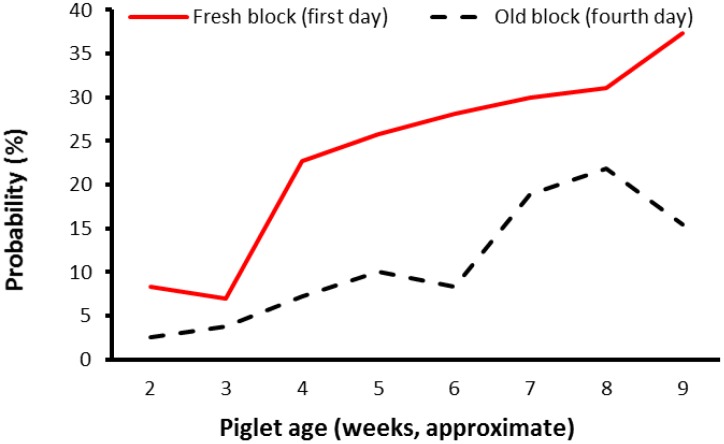
Likelihood that any pig(s) was/were performing oro-nasal contact with the blocks when observed, on the first compared to fourth day that blocks were available within weeks. The values shown are back-transformed means expressed as proportions. Pigs were weaned at about age 28 days.

**Figure 4 animals-07-00091-f004:**
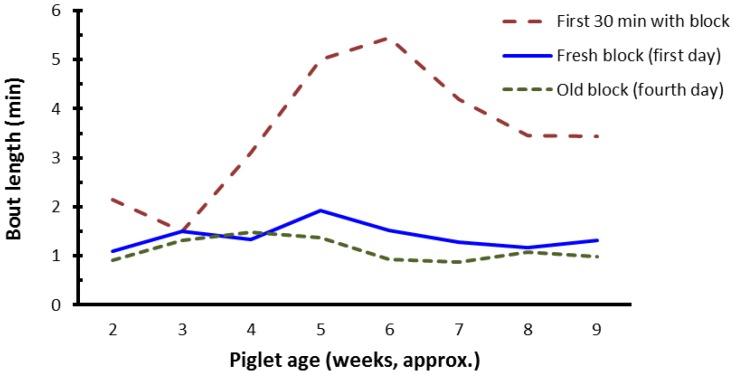
Mean bout length of oro-nasal contact with the shaped blocks by focal pigs in the experiment. The data presented are back-transformed mean values, showing bout duration in the first 30 min of placement of a fresh block, the first 24 h and the fourth day after the block was provided. Pigs were weaned at about age 28 days.

**Figure 5 animals-07-00091-f005:**
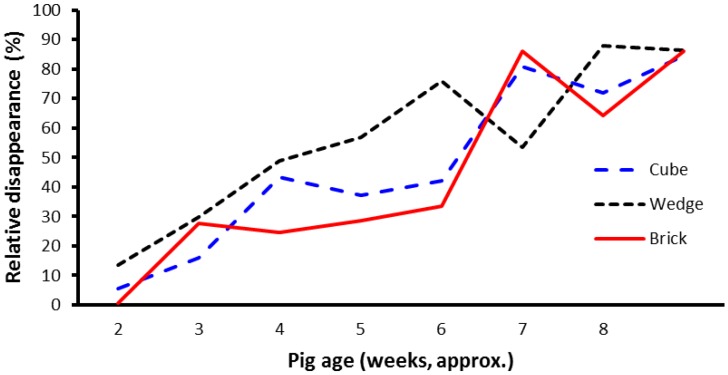
Relative disappearance of the shaped blocks over the course of the experiment. The values shown are mean percentages adjusted for number of pigs in the litter. Pigs were weaned at about age 28 days.

**Figure 6 animals-07-00091-f006:**
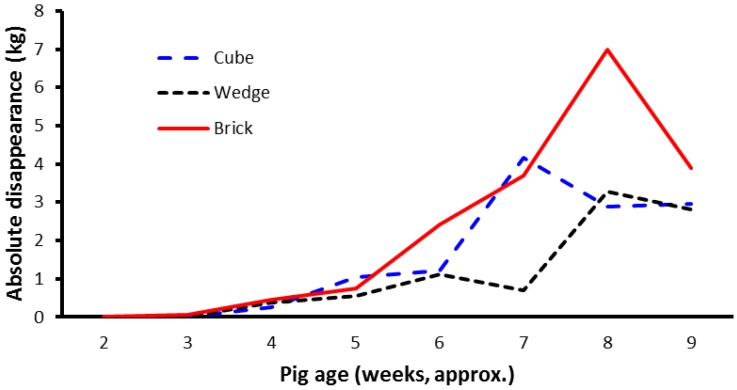
Absolute block disappearance of the shaped blocks over the course of the experiment. The data are predicted mean weights adjusted for number of pigs in the litter. Pigs were weaned at about age 28 days.

**Table 1 animals-07-00091-t001:** Dimensions and angles identified in Figure 1 for the different shaped blocks provided for sucker and weaner pigs in the experiment. Measurements shown are approximate.

Shape:	Cube				Brick				Wedge				
Unit	A	B	C	d	A	B	C	d	A	B	C	d	e
	cm	cm	cm	deg	cm	cm	cm	deg	cm	cm	cm	deg	deg
**Sucker**	8	8	8	90	8	16	8	90	8	8	11	90	45
**Weaner**	13	13	13	90	13	26	13	90	13	13	18	90	45

**Table 2 animals-07-00091-t002:** Number of litters in each block shape treatment per replicate (farrowing batch).

Shape:	Cube	Brick	Wedge
**Replicate 1**	3	2	3
**Replicate 2**	1	2	2
**Replicate 3**	2	2	2
**Total litters**	6	6	7

**Table 3 animals-07-00091-t003:** Live weight of pigs and group (litter) size details for each block shape treatment during the experiment. Week 1 refers to week 1 on experiment, which commenced when piglets were about 10 days old. In replicate 1, the groups remained on treatment for 8 weeks, whereas in replicates 2 and 3, the groups remained on treatment for 6 weeks. Values shown are means (±std dev).

Week of Trial	Age ^†^	Cube		Brick		Wedge	
	(Days)	Wt (kg)	N Pigs	Wt (kg)	N Pigs	Wt (kg)	N Pigs
1	11	3.49 (0.58)	10.5 (1.64)	4.16 (0.71)	9.5 (1.38)	3.64 (1.01)	11.0 (2.65)
2	18	5.48 (0.76)	10.5 (1.64)	5.92 (1.14)	9.5 (1.38)	5.49 (1.28)	11.0 (2.65)
3	25	7.55 (1.15)	10.5 (1.64)	8.29 (1.62)	9.5 (1.38)	7.61 (2.03)	10.6 (2.88)
4	32	9.33 (1.43)	9.8 (1.72)	9.85 (1.04)	9.5 (1.38)	9.40 (1.40)	10.4 (2.64)
5	39	10.89 (1.55)	9.7 (1.63)	11.52 (1.08)	9.5 (1.38)	11.09 (1.86)	10.4 (2.64)
6 ^‡^	46	13.61 (2.44)	11.0 (1.00)	13.32 (1.29)	10.5 (0.71)	12.43 (0.45)	12.3 (2.08)
7 ^‡^	53	15.69 (2.71)	11.0 (1.00)	15.42 (1.81)	10.5 (0.71)	14.95 (0.50)	12.0 (1.73)
8 ^‡^	60	18.03 (2.85)	11.0 (1.00)	17.59 (2.12)	10.5 (0.71)	17.31 (0.46)	12.0 (1.73)

^†^ approximate age of pigs at the commencement of the trial week; ^‡^ Data for weeks 6–8 of the trial are only for replicate 1.

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
