# Peer review of "Environmental Enrichment for Sucker and Weaner Pigs: The Effect of Enrichment Block Shape on the Behavioural Interaction by Pigs with the Blocks"

_animals, 2017, doi:10.3390/ani7120091_

Round 1

Reviewer 1 Report

Dear Editor and Authors,

I evaluated the manuscript entitled “Environmental enrichment for sucker and weaner pigs: the effect of enrichment block shape on the behavioural interaction by pigs with the blocks” and I found it relevant and overall well-written. However, I have some concerns with respect to methodology and data presentation, reported as specific comments below, which will need to be addressed:

INTRODUCTION

Lines 52-53: the existence of legislation requirements should be mentioned beside public expectations

MATERIAL AND METHODS

Line 111: was creep feeding provided as well? It is mentioned on line 319 but it should be reported here. Why wasn’t it supplied? Could this have increased block disappearance, especially during the last week of lactation? This aspect should be discussed.

Line 113: how many piglets/pen? Were some litters split? These pens appear to be just to small to host the entire litters for the entire trial. Which was the piglets BW at the beginning and at the end of the trial? The space allowance per piglet should be stated. Also, could the different number of piglets per block have biased the results? This aspect should be discussed.

Lines 103-114: were piglets castrated, tail docked or were their teeth clipped? Did they undergo any sanitary issue or other manipulation/treatment during the trial?

Line 118: remove “respectively”

Line 149: remove “consumption or”

Table 2: I am wondering why treatments were not balanced across replications. Can the Authors explain and comment on this? In which season of the year was each replication? Did they observe differences between replications? Was there any other factor (apart from the season) which could have biased the results across replications?

Line 170 (and throughout the manuscript): change “weight change of the block” to “block disappearance”

Line 178-179: “lines were left on”. Why not using just dim lights at night? I am afraid this could have disrupted piglets’ and sows’ rest, raising some evident welfare issues.

Lines 180-208: behavioural observations: it has been reported that the scan sampling technique is not the best for the study of behaviours lasting short times, such as drinking or interactions with the enrichment. For this purposes, in-continuous video observation (all-occurrences sampling) should be used. With this respect, Authors should explain their methodological choice (did they base on any previous paper or other literature?), or disclose any possible observational bias deriving from their choice. Besides, why were focal pigs used for the duration of interaction, but not for the frequency of interaction?

Lines 202-203: this information should be moved to the section “enrichment block shape treatment”

Line 219: how were means back-transformed?

RESULTS

General concern: at lines 159-163 it is reported that the second and third replicates ended at 46 days of age. Does this mean that data collected at days 53 and 60 in the graphs only refer to the first replicate? If so, this should be highlighted in the graphs and discussed also considering that the trend at this age appears to change with respect to younger ages. Also, Authors mention that the wedge block was daily destroyed at this age. Was it replaced every day? Or did pigs remain without enrichment material? Could this have reduced the interactions with the wedge block observed at age 53 and 60 (see Figures 2 and 3)?

I strongly recommend adding the number of pigs per treatment observed both to the “material and methods” section and to each of the the figures.

Figures 2-6: considering that days are approximate, that observations were not made on the same day (for example, the horizontal axis in the third graph of Figure 2 should read days 15 – 22 – 29 – 36 etc.) and that observations made on different days are compared (day 1 and day 4, see figures 3 and 4), I strongly recommend to re-label axis indicating the weeks of age, instead of the days.

Line 243: considerations on habituation to the block should be moved to the “discussion” section

Line 256-259: I suggest removing this sentence or adapting and moving it to the “discussion” section

Lines 285-287: change “weight change/reduction of the block” to “block disappearance”

Line 298: it is not necessary to repeat data here

Line 306: “nursing behaviour”

Line 311: I suggest removing “regardless of whether they were positioned shoulder-to-shoulder” since udder massage is not the only co-operative group behaviour

Line 317: change “the pigs grow” to “they”

Line 319: why was creep feed not supplied? Could this have increased the edible block consumption, biasing the estimates?

Line 326-327: this sentence is not clear to me. Also, considering the practical impossibility to provide new block every week under intensive farming conditions, would the Authors deem important also studying the block consumption (and the piglets' behaviour) until the block is finished?

Line 330: “it slowed down the rate of habituation”

Author Response

Reviewer 1

This paper addresses an important topic. It is generally very well written, with a sound experimental design, interesting results and a good discussion.I recommend the paper be published subject to a few very minor changes (see below).

Introduction

On Line 50, the authors point out that environmental enrichment (EE) is provided to zoo animals. This is true, but EE is also provided to companion, laboratory and farm animals, and perhaps the authors should mention this to give some more background on the topic.

Response: Good point from the reviewer. The text has been modified to reflect this point, with the following text added:

as well as companion, laboratory and farm animals

The full sentence now reads: Environmental enrichment is provided to captive zoo animals, as well as companion, laboratory and farm animals to assist in improving welfare by increasing the frequency and diversity of (natural) behaviours performed [1-3].

On Line 55 the authors say "critics suggest that social living per se does not satisfy pig welfare". I think this is an understatement as it is not only critics, but also compelling scientific evidence, that does indeed indicate that this is the case. I suggest the authors rephrase this sentence.

Response: We have re-worded the sentence to hopefully improve the emphasis along the lines suggested by the reviewer. The original sentence was:

Although most pigs are housed in groups enabling social interaction, whether positive or negative, critics suggest social living per se does not satisfy pig welfare requirements [6].

The new sentence is:

In order to maximise welfare outcomes, young pigs need to be housed in groups enabling social interaction, and with sufficient space and enrichment materials to stimulate normal exploratory behaviours [6].

Methods

Please give details on the precise location of the blocks particularly when used with sucking piglets. The authors say they were attached to the vertical bars between two farrowing pens, but the precise location in terms of the distance from the sows' head may be important in the case of very young piglets.

Response: The text has been modified to clarify the position of the blocks in the crate and weaner pen environments. The full paragraph from the M&M is as follows, with the added text shown in blue font.:

Each shaped block was drilled through using a 10 mm diameter drill bit, which enabled the block to be skewered on a 10 mm-thick metal rod (see Figure 1). Thus, the blocks could be fixed on the vertical rods within farrowing crates/weaner pens to facilitate observation via video cameras. In addition, the shaped blocks were raised 20 mm above the slatted floor via a spacer located under the block, through which the metal rod passed. In farrowing crates the attachment rod skewering blocks was positioned about 20-25 cm from the front and (creep-) side wall of the farrowing crate, and about 35-40 cm from the sow stall. In weaner pens the attachment rod was positioned centrally, about 25-30 cm from the rear wall. Blocks were positioned above slatted rather than solid flooring to facilitate attachment of the lower end of the vertical metal rod from below the floor.

Results

It would be interesting to know if there is any evidence that multiple piglets interacted with brick-shaped blocks. Even if the authors do not have quantitative data on this, including a comment on whether this was observed would be of interest to back up the discussion.

Response: Good suggestion by the review. I went back through the data set and summarized the following information from the approx. 30,000 observations collated from video by Jade Winfield. For the Cube, Brick and Wedge block-shapes, one pig was seen interacting with the block on 8.4, 10.1 and 8.1% of occasions, respectively. However, the proportion of observations in which more than 1 pig was interacting with the block was 6.4, 7.3 and 5.8% of occasions, respectively.

The following text has been added to the end of the first paragraph in the Results – section 3.1:

In total during the study, about 30,000 observations were collated from the video records. For the Cube, Brick and Wedge block-shapes, respectively, a single pig was recorded interacting with a block on 8.4, 10.1 and 8.1% of occasions, while 2 or more pigs interacted on 6.4, 7.3 and 5.8% of occasions. Thus, during the majority of observations (~85%) pigs did not interact with the blocks.

Discussion

The first hypothesis (ie that young piglets will use wedge-shaped blocks to a greater extent than other types of blocks) was not supported by the results, This is fine, but it would be useful if the authors could make an attempt at explaining why this could be the case.

It was found that young piglets did not interact much with blocks. The authors try to explain this based on the fact that their foraging drive may be low as the sow is still producing a significant amount of milk. This can be the case, but other possible explanations do exist. For example, maybe the location of the blocks was not adequate or, alternatively, none of the block types used in this study was attractive to very young piglets. I would suggest the authors widen up the discussion by including other possible explanations.

Response: Thank you for these thought-provoking comments. The enrichment block was placed within each farrowing crates just forward of the edge of the solid floor area comprising the heated creep area. Because the blocks were positioned on the same side as the creep, where piglets spend most of their time in crates, we are fairly certain they would have been aware of the blocks. The suggestion that the block material or shapes were not attractive (or had low relevance) is useful, and we have added a sentence regarding the possibility. In addition, a relevant point that we overlooked in the discussion was that weaning occurred around day 28 of age. Although sucker pigs probably have increasing motivation to obtain food (sow milk) as they grow, and especially once the sow reaches her peak milk production, the time span around which the increase in oro-nasal contact seemed to be elevated was followed soon after by weaning. Hence, we may have also recorded a response in increased motivation due to weaning.

The context of the added text within the paragraph is shown here:

A key finding from this experiment however was that there was relatively little oro-nasal contact with any of the blocks until the litters were between 18 to 25 days of age (see Figures 2 and 3). Alternatively, it is possible that at younger ages piglets perceived the type of material in the block, or the block shapes, to have low relevance. Nevertheless, younger pigs did not seem to be as ‘attracted’ to the blocks as older pigs. The finding may be associated with changes in sow milk production, or even with weaning per se which occurred on average around 28 days of age. Sows reach peak milk production by about day 21 of lactation [19], and thereafter milk output does not increase, whereas once piglets are weaned sow milk is no longer available and the weaner pigs need to consume an alternative form of feed. Regardless, after peak milk production is reached or weaning occurs, appetite would be expected to continue to increase as they grow.

A little further on in that paragraph:

In the farrowing/lactation environment, although creep feed was supplied, it was supplied in relatively small quantities each day and hence the main source of food for sucker pigs was the sow’s milk. Presumably, as lactation progresses towards weaning the gap between appetite of the sucker pigs and the amount of milk available from the sow increases, motivating piglets to perform increased levels of foraging behaviour.

Reviewer 2 Report

This paper addresses an important topic. It is generally very well written, with a sound experimental design, interesting results and a good discussion.I recommend the paper be published subject to a few very minor changes (see below).

Introduction

On Line 50, the authors point out that environmental enrichment (EE) is provided to zoo animals. This is true, but EE is also provided to companion, laboratory and farm animals, and perhaps the authors should mention this to give some more background on the topic.

On Line 55 the authors say "critics suggest that social living per se does not satisfy pig welfare". I think this is an understatement as it is not only critics, but also compelling scientific evidence, that does indeed indicate that this is the case. I suggest the authors rephrase this sentence.

Methods

Please give details on the precise location of the blocks particularly when used with sucking piglets. The authors say they were attached to the vertical bars between two farrowing pens, but the precise location in terms of the distance from the sows' head may be important in the case of very young piglets.

Results

It would be interesting to know if there is any evidence that multiple piglets interacted with brick-shaped blocks. Even if the authors do not have quantitative data on this, including a comment on whether this was observed would be of interest to back up the discussion.

Discussion

The first hypothesis (ie that young piglets will use wedge-shaped blocks to a greater extent than other types of blocks) was not supported by the results, This is fine, but it would be useful if the authors could make an attempt at explaining why this could be the case.

It was found that young piglets did not interact much with blocks. The authors try to explain this based on the fact that their foraging drive may be low as the sow is still producing a significant amount of milk. This can be the case, but other possible explanations do exist. For example, maybe the location of the blocks was not adequate or, alternatively, none of the block types used in this study was attractive to very young piglets. I would suggest the authors widen up the discussion by including other possible explanations.

Author Response

Reviewer 2

evaluated the manuscript entitled “Environmental enrichment for sucker and weaner pigs: the effect of enrichment block shape on the behavioural interaction by pigs with the blocks” and I found it relevant and overall well-written. However, I have some concerns with respect to methodology and data presentation, reported as specific comments below, which will need to be addressed:

INTRODUCTION

Lines 52-53: the existence of legislation requirements should be mentioned beside public expectations

Response: Correct. In some countries the provision of EE is also legislated for pigs. The following text has been added to the Introduction to include the point:

In some countries this is also a legislative requirement.

MATERIAL AND METHODS

Line 111: was creep feeding provided as well? It is mentioned on line 319 but it should be reported here. Why wasn’t it supplied? Could this have increased block disappearance, especially during the last week of lactation? This aspect should be discussed.

Response. The question about creep feed is important, so many thanks for raising it. As a result I went back to check this with the pig unit manager. He indicated that from 7 days of age he provides a small amount of (pelleted) creep feed for litters daily, at a rate of about 1 handful per day which he places on the solid floor of the creep area. He does not provide a specific creep feeder. The text has thus been modified in the M&M and then the Discussion.

M&M:

A small amount (handful per crate) of creep feed was provided daily on the floor of the creep area from day 7, and water was available ad libitum via a piglet nipple drinker.

Discussion:

In the farrowing/lactation environment, although creep feed was not supplied, it was supplied in relatively small quantities each day and hence the main only source of food for sucker pigs was the sow’s milk.

Line 113: how many piglets/pen? Were some litters split? These pens appear to be just to small to host the entire litters for the entire trial. Which was the piglets BW at the beginning and at the end of the trial? The space allowance per piglet should be stated. Also, could the different number of piglets per block have biased the results? This aspect should be discussed.

Responses: There are many relevant questions and statements here from the reviewer. The text has been separated accordingly, and answers are shown below:

how many piglets/pen? Which was the piglets BW at the beginning and at the end of the trial? The mean number of pigs per crate/pen, and mean live weights, have been added at the start of the Results section for key time points in the trial. The data are presented as a (new) Table (= Table 3).

Were some litters split? Litters were not split – they were maintained as they were, throughout the trial. Adjustment of litter size by cross-fostering only occurred for 2 of the 19 litters in the trial. Sow O67 gave birth to 3 piglets, and 5 piglets were fostered on to this sow from a non-experimental litter. In another litter (Sow O71), only 6 piglets were born alive and 2 of these died soon after birth. 8 piglets were fostered onto this sow. Fostering occurred within 3 days of parturition, and was conducted by the pig unit manager as per requirement to enhance piglet survival. Where the number of piglets per litter declined during the trial, this was mainly due to mortality.

These pens appear to be just to small to host the entire litters for the entire trial. The space allowance per piglet should be stated. The number and weight of weaner pigs per weaner pen were monitored weekly, as is required under our Animal Ethics Committee obligation. The maximum litter size in the weaner pens was 13 pigs. Two litters in replicate 1 had 13 pigs at the end of the trial. The average pig weight in the 2 groups was about 17 kg / pig at the completion of the trial (~60 days of age). Under the Australian Pig Welfare Code of Practice, 17 kg live weight pigs require a minimum space allowance of 0.20 m2/pig. Thus 13 pigs x 0.20 m2 = 2.6 m2; the weaner pens measured 2.82 m2. The heaviest pigs averaged about 20.5 kg at the conclusion of the trial (in Rep 1), and there were 12 pigs in the group. In no case did we exceed our obligated space allowance under the Australian Pig Welfare Code. In replicates 2 and 3, the trial ended when the pigs were about 7 weeks old, and thus the pigs were much smaller / lighter in weight.

Also, could the different number of piglets per block have biased the results? This aspect should be discussed. A limitation of our trial was that we had a relatively small number of litters (overall N = 19 litters; Cube = 6 litters, Brick = 6 litters, Wedge = 7 litters). Our pig unit is relatively small (48 sows) and we batch farrow at about 8 week intervals. Also, this trial was conducted as an Honours Student Project, so the number of sows and litters was limited by herd size, farrowing batch frequency and time constraints on the student’s semester timeline. There was no difference between the block-shape treatments and average litter size in the treatment groups. For example, at week 5 of the trial (pig age about 46 days) the ave litter sizes were 9.7, 9.5 and 10.4 pigs for the Cube, Brick and Wedge shape treatments, respectively. The mean (and std dev) number of piglets per litter at that time for the 19 litters were 9.9 pigs (1.94). Nevertheless, the point made by the reviewer is quite valid, and the text has been modified in the M&M and Results to provide more detail for the reader, and a comment has been added to the Discussion acknowledging this limitation.

Modifications to text in the M&M section 2.1:

The experiment was conducted at the University of Sydney piggery, Camden, under approval of the University of Sydney Animal Ethics Committee (approval number: 2016/968). A total of 197 piglets from 19 litters were used. The dams were crossbred Large White-Landrace sows, and litter size (total born) at farrowing ranged from 3-19 piglets. Mean (± std dev) sow parity number was 2.9  ± 1.59 litters, and born alive per litter was 11.5 ± 3.67 piglets. At about 24-36 h post partum, piglets were weighed and underwent ear-notching, tail docking (last one-third of tail) and teeth clipping in line with the husbandry practices on the farm. Piglets also received an intramuscular iron injection.

In the Results section, the following text and Table (Table 3) have been added to help address the points made by Reviewer #2.

3.1. Animal numbers, weights and space allowance during the experiment

The experiment commenced when the different shaped enrichment blocks were introduced to the litters in the farrowing crates. As explained above in section 2.3, for replicates 2 and 3 the trial period ended when the pigs were approaching 7 weeks old. Details of litter size and live weight of the pigs as they progressed through the trial are presented in Table 3. It should be noted that in Table 3 the full data set corresponding to all 19 litters (3 replicates) is limited to weeks 1-5 of the trial. The space allowance at week 8 on trial for the group with the heaviest mean weight (20.46 kg / pig, Rep. 1, litter 8) and largest group at week 8 (n = 13 pigs, Rep. 1, litters 3 & 6, 16.92 & 17.18 kg, respectively) were always within the Australian Pig Welfare Code minimum space allowance requirements. The minimum space allowance for 21 kg pigs is 0.23 m2.

Table 3. Live weight of pigs and group (litter) size details for each block shape treatment during the experiment. Week 1 refers to week 1 on experiment, which commenced when piglets were about 10 days old. In replicate 1, the groups remained on treatment for 8 weeks, where as in replicates 2 and 3, the groups remained on treatment for 6 weeks. Values shown are means (± std dev).

Wk of

Age †

Cube

Brick

Wedge

trial

(days)

Wt (kg)

N pigs

Wt (kg)

N pigs

Wt (kg)

N pigs

1

11

3.49 (0.58)

10.5 (1.64)

4.16 (0.71)

9.5 (1.38)

3.64 (1.01)

11.0 (2.65)

2

18

5.48 (0.76)

10.5 (1.64)

5.92 (1.14)

9.5 (1.38)

5.49 (1.28)

11.0 (2.65)

3

25

7.55 (1.15)

10.5 (1.64)

8.29 (1.62)

9.5 (1.38)

7.61 (2.03)

10.6 (2.88)

4

32

9.33 (1.43)

9.8 (1.72)

9.85 (1.04)

9.5 (1.38)

9.40 (1.40)

10.4 (2.64)

5

39

10.89 (1.55)

9.7 (1.63)

11.52 (1.08)

9.5 (1.38)

11.09 (1.86)

10.4 (2.64)

6 ‡

46

13.61 (2.44)

11.0 (1.00)

13.32 (1.29)

10.5 (0.71)

12.43 (0.45)

12.3 (2.08)

7 ‡

53

15.69 (2.71)

11.0 (1.00)

15.42 (1.81)

10.5 (0.71)

14.95 (0.50)

12.0 (1.73)

8 ‡

60

18.03 (2.85)

11.0 (1.00)

17.59 (2.12)

10.5 (0.71)

17.31 (0.46)

12.0 (1.73)

† : approximate age of pigs at the commencement of the trial week

‡ : Data for weeks 6-8 of the trial are only for replicate 1

Additional comment added to the end of the Discussion:

Similarly, we acknowledge that this was a relatively small study and a larger study with more litters is justified. Nevertheless, with more than 30,000 observed time-points included for this observational study, useful information is presented to assist in the design of enrichment blocks for smaller pigs.

Lines 103-114: were piglets castrated, tail docked or were their teeth clipped? Did they undergo any sanitary issue or other manipulation/treatment during the trial?

Response: In Australia, almost no pig farms castrate male pigs (we are not aware of any farms, anyway). In fact, the issue to avoid castration is identified under the Australian pig welfare code, that is, that castration should not be performed. Thus, no piglets in our study were castrated. Piglets were tail-docked (last one-third of the tail only), teeth clipped (if necessary) and ear notched (for individual identification). Relevant information has been added to the text under M&M section 2.1:

At about 24-36 h post partum, piglets were weighed and underwent ear-notching, tail docking (last one-third of tail) and teeth clipping in line with the husbandry practices on the farm. Piglets also received an intramuscular iron injection.

Line 118: remove “respectively” Response: Done

Line 149: remove “consumption or” Response: Done

Table 2: I am wondering why treatments were not balanced across replications. Can the Authors explain and comment on this? In which season of the year was each replication? Did they observe differences between replications? Was there any other factor (apart from the season) which could have biased the results across replications?

Response: As explained above, our pig unit is a small unit and although we aim to farrow a certain number of sows (8) per batch, this does not always eventuate. In replicates 2 and 3 only 5 and 7 sows, respectively, were available. In replicate 3 one of the 7 sows became ill around farrowing, and all her piglets were weaned off and transferred to 2 other sows. Fortunately, one of our trial sows only had 3 piglets, so she was able to take 6 piglets from the sick sow. Also, we only have 8 farrowing crates, and with 3 treatments the experimental design will likely be unbalanced. However, our statistician (Dr Evelyn Hall) analysed the data using Generalised Linear Mixed Models which were appropriate for the experimental design and data set, which ensured the issue of imbalance was not a problem.

The 3 batches (replicates) of sows farrowed in (1) early April, (2) late May, and (3) late June (southern hemisphere). The farrowing room had been recently renovated, and it is well insulated and temperature is thermostatically controlled which allows us to maintain optimal temperatures for the piglets.

We do not believe that additional changes to the MSS are required on this point.

Line 170 (and throughout the manuscript): change “weight change of the block” to “block disappearance”

Response: Done, including on the relevant figures.

Line 178-179: “lines were left on”. Why not using just dim lights at night? I am afraid this could have disrupted piglets’ and sows’ rest, raising some evident welfare issues.

Response: We routinely leave the back-ground lights on in our farrowing and weaner pig rooms, especially for security reasons at night. From our experience this does not seem to impact the sows or piglets. In some of my (GMC) earlier research on video recording pre-farrowing behaviour by sows in crates and pens (conducted in the late 1980’s), I actually compared the time budgets of behaviour of farrowing / lactating sows and piglets under different lighting conditions, including red light, yellow light, (existing) fluro-tube white light and no light (IR light) at night, with the use of early models of IR video cameras. Although I did not publish my data from that work, I conducted the study to determine whether leaving the background lights on at night when I video-recorded the pigs would alter behavioural time budgets. The behaviour of the pigs did not differ, regardless of the type of lighting or illumination level. In addition, as explained below, each crate / pen had a heater that emitted (bright) light, and we needed to try to even-out the light to improve the clarity of pigs near the blocks to ensure they could be individually identified on the video record. The following text has been added to the MSS to improve the description:

This was necessary to help avoid over-exposure of pigs on the video record, whilst also improving illumination of the blocks. The blocks were relatively dark in colour and the pig heaters emitted bright (visible) light that could cause over-exposure within the field of view reducing the observer’s ability to identify focal pigs.

Lines 180-208: behavioural observations: it has been reported that the scan sampling technique is not the best for the study of behaviours lasting short times, such as drinking or interactions with the enrichment. For this purposes, in-continuous video observation (all-occurrences sampling) should be used. With this respect, Authors should explain their methodological choice (did they base on any previous paper or other literature?), or disclose any possible observational bias deriving from their choice. Besides, why were focal pigs used for the duration of interaction, but not for the frequency of interaction?

Response: The reviewer makes a valid point here. However, a number of researchers have modelled the comparison of continuous recording against various rates of scan sampling, and reported that as the number of point samples increases, the correlation with the continuous observation method becomes progressively stronger. For the present study, Jade (the Honours student) collated data on the number of pigs at the blocks during more than 30,000 time points to estimate the frequency of oro-nasal interaction with the blocks. In addition, to estimate bout length of oro-nasal interactions, she also recorded more frequently (per minute) for set periods of 5 consecutive 1-min time points using 4 focal pigs per group. Overall, this was a massive undertaking, and in our opinion has generated an impressive data set that would (practically speaking) not be achieved via someone performing continuous observation. One paper that we referred to in preparation for this trial was by Daigle & Siegford (2014) Behav Proc v103 pp 58-66.

We do not believe it is necessary to modify the MSS, but should the Editor request it we will make any necessary changes recommended.

Lines 202-203: this information should be moved to the section “enrichment block shape treatment”

Response: Done – now stated in the initial paragraph in that section

Line 219: how were means back-transformed?

Response:– the statistical programmes and, or the statistician calculate the back-transformed means via the appropriate, prescribed method.

RESULTS

General concern: at lines 159-163 it is reported that the second and third replicates ended at 46 days of age. Does this mean that data collected at days 53 and 60 in the graphs only refer to the first replicate? If so, this should be highlighted in the graphs and discussed also considering that the trend at this age appears to change with respect to younger ages. Also, Authors mention that the wedge block was daily destroyed at this age. Was it replaced every day? Or did pigs remain without enrichment material? Could this have reduced the interactions with the wedge block observed at age 53 and 60 (see Figures 2 and 3)?

Response: The text has been modified in a number of places reinforcing that replicates 2 and 3 ended sooner than replicate 1. When a (wedge) block was destroyed, it was replaced the next morning with a new block of the appropriate shape, even if this was every day. These events were un-planned and as a result we needed to make some decisions on the run in replicate 1, and then decided to avoid the problem altogether by ending reps 2 & 3 earlier than rep 1. We agree this was problematic and potentially introduced (un-necessary) variance to the data-set. The following text has been added towards the end of the Discussion to recognise concerns over this limitation.

A particular issue occurred in the last few weeks of replicate 1, with the wedge-shaped blocks being prone to complete destruction by the weaner pigs. Accordingly, new blocks were replaced in the pens, sometimes daily. To avoid this problem in replicates 2 and 3 we decided to conclude the subsequent replicates sooner. Thus, behavioural responses indicative of habituation in the wedge-shaped treatment in weeks 6-8 on experiment refer only to responses measured in replicate 1. Therefore, caution is required since disappearance rate of the wedge block may be confounded by block-age and not an effect of the block treatment per se, although the wedge shape and lower mass probably contributed to the pigs’ ability to (learn to) destroy that block shape faster.

I strongly recommend adding the number of pigs per treatment observed both to the “material and methods” section and to each of the the figures.

Response: Done (see Table 3)

Figures 2-6: considering that days are approximate, that observations were not made on the same day (for example, the horizontal axis in the third graph of Figure 2 should read days 15 – 22 – 29 – 36 etc.) and that observations made on different days are compared (day 1 and day 4, see figures 3 and 4), I strongly recommend to re-label axis indicating the weeks of age, instead of the days.

Response: This is a good point by the reviewer. The X-axis of relevant figures have been modified to show (approx) weeks of age rather than days.

Line 243: considerations on habituation to the block should be moved to the “discussion” section

Response: The sentence has been omitted from the Results, and a small modification made to the text in the Discussion (see below):

While the frequency of pigs contacting the blocks increased with pig age (see Figure 3), within weeks the performance of oro-nasal behaviour directed at the blocks was also significantly greater in the first 24 h after fresh blocks were provided, compared to 4 days later and, is suggestive of habituation to the blocks. Habituation to the blocks however, did not seem to occur long-term, probably as the blocks were replaced weekly.

Line 256-259: I suggest removing this sentence or adapting and moving it to the “discussion” section

Response: Done

The data in Figure 4 may also be useful to estimate whether pigs had habituated to the blocks within weeks of age, by visually comparing the duration of oro-nasal contact by pigs with the blocks between the initial 30 min (1st 30 min), the first 24 h (Day 1) and for a 24-h period 4 days after the block entered the enclosure (Day 4).

Lines 285-287: change “weight change/reduction of the block” to “block disappearance”

Response: Done

Line 298: it is not necessary to repeat data here

Response: Done – percent values have been omitted from the text.

Line 306: “nursing behaviour”

Response: Done

Line 311: I suggest removing “regardless of whether they were positioned shoulder-to-shoulder” since udder massage is not the only co-operative group behaviour

Response: Done

Line 317: change “the pigs grow” to “they”

Response: Done

Line 319: why was creep feed not supplied? Could this have increased the edible block consumption, biasing the estimates?

Response: As mentioned above, we later found out that a small amount of creep feed was given to each litter daily by the pig unit manager. The text has been modified as follows:

M&M:

A small amount (handful per crate) of creep feed was provided daily on the floor of the creep area from day 7, and water was available ad libitum via a piglet nipple drinker.

Discussion:

In the farrowing/lactation environment, although creep feed was not supplied, it was supplied in relatively small quantities each day and hence the main only source of food for sucker pigs was the sow’s milk.

Line 326-327: this sentence is not clear to me. Also, considering the practical impossibility to provide new block every week under intensive farming conditions, would the Authors deem important also studying the block consumption (and the piglets' behaviour) until the block is finished?

Response: We agree, the sentence could be worded better to improve the logic. Also, the reviewer’s suggestion that a study should investigate the “use” of these blocks without replacement is a good idea, especially in a commercial situation. However, as explained earlier, the current experiment was a student honours project. Nevertheless, part of the rationale for conducting the present experiment was to determine whether there was a preferred block shape for smaller pigs. That information was subsequently used to inform a much larger industry project which was completed only recently. The sentence has been replaced with the following new text:

Potentially, the fact that blocks were replaced weekly may have slowed habituation.

Line 330: “it slowed down the rate of habituation”

Response: Done

Round 2

Reviewer 1 Report

The manuscript has significantly improved after the revision and I recommend its publication in the present form.

Author Response

The aim of this experiment was to investigate the effect of different block shapes on the pigs interaction with the blocks. While blocks were similar in size and weight volume was not considered and as such was not incorporated into the hypothesis that was tested. Blocks were cut to shape using larger blocks that were previously designed for use by sows, these measured: 26cm x 26cm x 26cm. These blocks are specially formulated and not available commercially at the time of submission. The sizes were chosen such that they were appropriate relative to the size of the piglets. These details are contained in the materials and methods section of the manuscript.

Volume was not our focus and not incorporated into the hypothesis that we tested. The methods we used were entirely appropriate to test our hypothesis that centred around block shape. We have not modified the text in the MSS at this time, but will happily abide by the Editor's decision to add text on the identified point, as required, to improve the MSS.